# Basophil Characteristics as a Marker of the Pathogenesis of Chronic Spontaneous Urticaria in Relation to the Coagulation and Complement Systems

**DOI:** 10.3390/ijms241210320

**Published:** 2023-06-19

**Authors:** Yuhki Yanase, Daiki Matsubara, Shunsuke Takahagi, Akio Tanaka, Koichiro Ozawa, Michihiro Hide

**Affiliations:** 1Department of Pharmacotherapy, Institute of Biomedical and Health Sciences, Hiroshima University, Hiroshima 739-0046, Japan; yyanase@hiroshima-u.ac.jp (Y.Y.);; 2Department of Dermatology, Institute of Biomedical and Health Sciences, Hiroshima University, Hiroshima 739-0046, Japan; 3Hiroshima City Hiroshima Citizens Hospital, Hiroshima 730-8518, Japan

**Keywords:** chronic spontaneous urticaria (CSU), basophil, mast cell, histamine, biomarker, coagulation, complement, autoantibody

## Abstract

Chronic spontaneous urticaria (CSU) is a common skin disorder characterized by daily or almost daily recurring skin edema and flare with itch and pruritus anywhere on the body for more than 6 weeks. Although basophil- and mast cell-released inflammatory mediators, such as histamine, play important roles in the pathogenesis of CSU, the detailed underlying mechanism is not clear. Since several auto-antibodies, IgGs which recognize IgE or the high-affinity IgE receptor (FcεRI) and IgEs against other self-antigens, are detected in CSU patients, they are considered to activate both mast cells in the skin and basophils circulating in the blood. In addition, we and other groups demonstrated that the coagulation and complement system also contribute to the development of urticaria. Here, we summarized the behaviors, markers and targets of basophils in relation to the coagulation–complement system, and for the treatment of CSU.

## 1. Pathogenesis of CSU

Urticaria is a common skin disease characterized by repetitive and transient appearance of skin edema and flare, mostly with itch, anywhere on the body [1]. It is classified based on the time from onset; acute urticaria with a course less than 6 weeks, and chronic urticaria lasting for 6 weeks or longer [1]. Lifetime prevalence of chronic urticaria is considered around 4%, with a substantial variation of time point prevalence among regions of the globe from 0.5% in Europe to 1.4% in Asia [2]. Chronic urticaria is further divided into chronic spontaneous urticaria (CSU), also called chronic idiopathic urticaria (CIU), and chronic inducible urticaria (CIndU). CSU is characterized by daily or almost daily recurring skin edema and flare without an apparent trigger for each emergence of eruption, whereas CIndU is characterized by the occurrence of wheals in response to specific stimuli, such as temperature, light and mechanical stress. Considerable evidence highlights the major and critical roles of mast cells in the skin and basophils in the skin or peripheral blood. However, the exact mechanism by which mast cells and basophils are activated in the skin and peripheral blood in patients with CSU is not yet clear. Approximately 40% of patients with CSU have IgG autoantibodies against IgE antibodies and/or the high-affinity IgE receptors (FcεRIs) (type IIb autoimmune) [3,4,5] (Figure 1). Moreover, IgE autoantibodies against several self-molecules, such as Interleukin (IL)-24, thyroid peroxidase (TPO), eosinophil peroxidase (EPO), eosinophilic cationic protein (ECP), double-strand deoxyribonucleic acid (dsDNA) and tissue factor (TF), which induce type I autoimmune CSU, have also been detected in certain populations of patients with CSU (type I autoallergy) [6,7,8,9,10,11] (Figure 1). 

A high efficacy of anti-IgE antibodies, such as omalizumab and ligelizumab, suggests the critical role of IgE antibodies in the serum of the pathogenesis of CSU [5]. In addition, the effectiveness of H1 anti-histamines for the treatment of CSU supports the importance of histamine, which is stored and released from skin mast cells and/or peripheral blood basophils, as well [12,13]. However, histamine release activity of such autoantibodies in type IIb autoimmune and autoantigens via auto-IgE in type I autoallergy CSUs is not very potent and does not change in the body of patients in accordance with their clinical symptoms. Moreover, these autoimmunities are not apparent in all patients with CSU. Therefore, the pathogenesis of CSU by autoantibody-independent mechanisms should also be considered in order to understand the whole picture of CSU. Recently, the increase of various substances and cytokines, such as substance P (SP), C-reactive protein (CRP), tumor necrosis factor (TNF)α, IL-1β, -6, -17, -31, -33, and transforming growth factor (TGF)β, has been reported in the plasma of patients with CSU [1,2,3,4,5,6,7,8,9,10,11,12,13,14,15]. In addition, coagulation/fibrinolysis factors in the plasma of CSU patients has also been reported, whereas the levels of IL-35 and vitamin D in the plasma of patients with CSU were decreased [16,17]. Moreover, various medications that target cytokines and receptors, such as IL-4, IL-5, IL-13, thymic stromal lymphopoietin (TSLP), complement 5a (C5a), C5a receptor (C5aR), sialic acid-binding immunoglobulin-like lectins 8 (siglec-8) and chemoattractant receptor-homologous molecule expressed on T-helper type 2 cells (CRTH2), are being investigated in clinical trials for the treatment of CSU. These findings suggest that the pathogenesis of CSU is much more complicated than previously understood [17,18,19,20].

## 2. Characteristics of Basophils and Mast Cells

It is well known that basophils and mast cells are important cells which contribute to the induction of inflammatory and/or allergic responses, especially type I allergy. Both human basophils and mast cells develop from hematopoietic stem cells into precursor cells of each lineage in the bone marrow. They share several characteristics, such as the presence of secretory granules showing metachromasia and the expression of FcεRI [21,22]. Cross-linkage of IgE antibodies bound to FcεRIs by a multivalent antigen, also called allergen, induce the phosphorylation of immunoreceptor tyrosine-based activation motif (ITAM) domain of FcεRI, resulting in the activation of tyrosine kinase, such as Lyn, spleen tyrosine kinase (Syk) and Bruton’s tyrosine Kinase (Btk), in both basophils and mast cells (Figure 1). Activation of these molecules then activate the functions of multiple intracellular molecules, such as protein lipase C (PLC), phosphoinositide 3-kinase (PI3K), mitogen-activated protein (MAP) kinases, protein kinase C (PKC), and increase Ca^2+^ concentration in cytoplasm, resulting in the release of preformed chemical mediators, such as histamine, protease and platelet-activating factor (PAF), stored in secretory granules. Basophils and mast cells also release newly synthesized inflammatory lipid mediators, including arachidonic acid metabolites, such as leukotriene (LT) and prostaglandin (PG), and inflammatory cytokines, such as IL-4 and IL-13 [21,22,23].

However, the process for maturation of basophils and mast cells are largely different. Precursor cells of basophil become mature basophils in response to IL-3 in the bone marrow and then circulate in the blood, accounting for only less than 1% of peripheral blood leukocytes [24,25]. Basophils may migrate into tissues with inflammation out of blood vessels. On the other hand, precursor cells of mast cells, that develop in the bone marrow, migrate into tissues, and mature in response to stem cell factor (SCF) under the influence of surrounding tissues. Human mast cells are divided into two groups, MC_TC_ and MC_T_, according to the expression of proteases, tryptase and chymase [26]. MC_TC_, expressing tryptase and chymase, resides in connective tissues, such as skin, and peritoneal cavity. MC_T_, only expressing tryptase, resides in mucosal tissues. Although MC_TC_ expresses C5aR on their surface, MC_T_ do not have C5aR [26]. The increase of serum tryptase can be taken as a marker of mast cell activation, because basophils do not release tryptase [21,22]. In addition, the lifetime of circulating basophils is only a few days (around 3 days), whereas it tends to be a few months or longer for tissue-resident mast cells [21,22]. 

Since basophils are a major source of IL-4, as well as in peripheral blood mononuclear cells, basophils are considered to play critical roles in type 2 inflammations and IgE antibody-associated allergic disorders (type I allergy), such as urticaria, asthma, pollen allergy, food allergy, anaphylactic shock and atopic dermatitis (AD) [21,22]. In addition, basophils express the IL-3 receptor (IL-3R) on the surface of cells, which play important roles for the enhancement of antigen-IgE antibody interaction-induced histamine release, as well as the development and survival of basophils [27]. 

Human skin mast cells, but not peripheral basophils, express mas-related G-protein coupled receptor member X2 (MRGPRX2), which is activated by various molecules, such as neuropeptides, including SP [28]. However, a recent study reported that basophils also quickly express MRGPRX2 on the surface of cells after FcεRI-dependent activation [29]. As inhibitory molecules of basophils and mast cells, we found adenosine, a metabolite of adenosine triphosphate (ATP), suppressed histamine release from human peripheral blood basophils and skin mast cells via A2a and A2b adenosine receptors, respectively [30]. Therefore, complement factors, proteases, neuropeptides and adenosine may also be critical regulators of peripheral basophils and skin mast cells in the pathogenesis of CSU. 

## 3. Basophil-Related Molecules in Plasma and on Cell Surface of Patients with CSU

Several basophil characteristics and their behavior in CSU patients are summarized in Table 1. In a group of patients, the concentration of histamine in the plasma of patients with CSU was significantly higher than that of healthy controls (Table 1) [31]. Moreover, spontaneous histamine release from basophils without stimulation in patients with CSU was increased, compared to that in healthy donors (Table 1) [31,32]. Nevertheless, the degree of spontaneous release of histamine was not correlated with total amount of histamine in the whole blood of patients with CSU [31], suggesting that the percent release of histamine from individual basophils of patients with CSU is higher than that of healthy control. Levels of IgE antibodies in serum are also increased in patients with CSU (Table 1) [32,33]. Moreover, IgE levels in serum are proportional to the levels of FcεRI expression on the surface of human peripheral basophils [33] (Table 1). We previously reported that a high concentration of IgE antibodies (>1 μM) activate IgE-depleted human peripheral basophils, resulting in the release of histamine [33]. As described above, the lifetime of basophils is only a few days. Therefore, a substantial population of basophils may be differentiated on a daily basis in the bone marrow and exposed to a high concentration of IgE antibodies in the peripheral blood and/or the bone marrow. That may result in the continuous release of histamine from newly developed peripheral basophils every day, and therefore, the elevation of histamine concentration in peripheral blood. Moreover, IgE-induced basophil activation and histamine release are enhanced in the presence of IL-3 [33]. In addition, the depletion of IgE antibodies in serum by anti-IgE antibodies, such as omalizumab and ligelizumab, decreases FcεRIs expressed on the surface of basophils by 78% in 10 days, together with a decrease in the severity of CSU [34]. Therefore, a high IgE concentration and high levels of FcεRI expression on the surface of basophils could be good markers of CSU. Paradoxically, a population of patients with CSU express a low level of FcεRI on basophils and is refractory to omalizumab [5]. Further studies on the relationship between high-IgE antibody concentrations in plasma and the expression level of FcεRI on the surface of peripheral basophils may unveil the detailed role of basophils in the pathogenesis of CSU, and enable us to develop more useful drugs for refractory CSU. 

In line with previous reports, the ratio of basophils in leukocytes and the number of basophils in the peripheral blood of patients with CSU is also significantly low compared to healthy donors and patients with AD (Table 1) [31,33,34,35]. Several reports demonstrated that basophils migrate and accumulate in urticarial lesions [36,37]. Migration of activated basophils to the dermis may decrease the number of basophils in peripheral blood. 

We reported that the expression levels of CD63 and CD69, degranulation markers, and CD203c, an activation marker on the surface of peripheral basophils of patients with CSU, were not significantly increased (Table 1) [31]. However, Vasagar et al. reported that the elevation of CD63, but not CD69 and CD203c, on the surface of basophils [38]. Mizuno et al. also reported that the expression of CD63 and CD203c on the surface of basophils are elevated compared to healthy donors [39]. Expression levels of degranulation/activation markers of basophils may change during cell preparation and timing of blood collection due to the migration of activated basophils to the dermis. 

The ratio of non-/low-responders, whose basophils in peripheral blood are not activated by IgE-FcεRI-dependent signals, tends to increase in patients with CSU compared to healthy donors and those with other allergic diseases, such as AD (Table 1) [31,34]. However, the reason why the number of no- or low- responsive basophils are increased in the patients with CSU remains unclear. Moreover, how basophils in patients of low- and non-responders release histamine is uncertain. We recently demonstrated that basophils of low- and non-responder CSU patients, whose basophils release low or no amounts of histamine in response to anti-IgE (FcεRI-dependent activation), maintain the capacity to release histamine in response to IgE-independent stimuli, such as C5a and N-Formylmethionyl-leucyl-phenylalanine (fMLP) [31,40]. Moreover, histamine release in response to C5a from basophils of patients with CSU was similar to or even higher than that of healthy donors (Table 1). Of note, a degree of histamine release from basophils in response to anti-IgE and, that to C5a in CSU responders, whose basophils can release normal amount of histamine in response to anti-IgE, tended to be negatively correlated, whereas those in healthy donors showed a tendency of positive correlation [31]. Therefore, IgE-FcεRI-independent signal transduction, such as C5a-C5aR interaction induced activation, may be a main pathway instead of IgE-dependent activation in the low/non-responders. Although basophils express C3aR on their surface, C3a did not induce histamine release from basophils unexpectedly [40]. C3a-C3aR interaction may contribute to the migration of basophils to lesions of urticaria.

**Table 1 ijms-24-10320-t001:** Changes of behavior of peripheral basophils in CSU.

Basophil-Related Conditions	Changes in CSU Patients
Plasma histamine level	Increase [31]
Spontaneous histamine release	Increase [31,32]
Plasma IgE level	Increase [33,34]
Expression level of FcεRI	Increase [33,34,39]
Reaction to anti-IgE antibody medication (omalizumab)	Quick (few days) [34,41,42]
Ratio of basophils in leukocytes	Decrease [31,33,34,35]
Activation marker (CD203c, CD69 and CD63)	No change or increase [33,38,39]
Migration to dermis	Increase [36,37]
Non- or low-responder	Tend to increase [31,34]
Reaction to C5a and fMLP	No change or increase [31,43]

## 4. Coagulation System, Protease Receptors and Complement System

Two major pathways, the intrinsic and the extrinsic coagulation pathways, are involved in the physiological blood coagulation. The intrinsic coagulation pathway is activated by the exposure of coagulation factors in the blood to collagen because of the abrasion of vascular endothelial cells from collagen. Coagulation factor XII is firstly activated and the coagulation cascade is proceeded, resulting in the formation of a coagulation clot. On the other hand, the extrinsic coagulation pathway is activated by the exposure of coagulation factors in the blood to TF, which is expressed on tissues out of blood vessel. Recently, the activation of the extrinsic coagulation cascade-producing active forms of the coagulation factors has been revealed in patients with CSU. Figure 2 shows a part of the extrinsic coagulation pathway. When a small amount of VIIa, the active form VII, binds to TF on vascular endothelial cells in the presence of calcium ion (Ca^2+^) and phosphatidylserine (PS), the extrinsic coagulation cascade is activated and produces active form of coagulation factors, such as coagulation factor Xa and thrombin (Factor IIa). Since activated basophils express PS on their surface, basophils may contribute to this step. The IIa then converts fibrinogen (I) to fibrin (Ia) which form fibrin polymers as a blood clot. Of note, heparin, which activates anti-thrombin, improved symptoms in cases of CSU [44]. Moreover, warfarin which inhibits the production of coagulation factors, such as coagulation factor VII, X, IX and II, has also been suggested to be effective for the treatment of CSU [44]. These findings imply that the activation of the extrinsic coagulation cascade and subsequently produced active form of coagulation factors play important roles as the trigger of CSU. PF1+2 is generated when factor Xa changes prothrombin (II) to factor IIa, followed by the generation of Ia polymers (coagulation clot) from factor I. D-dimers and fibrin/fibrinogen degradation products (FDPs) are produced in the process of fibrinolysis: the cleavage of fibrin polymers by plasmin (Figure 2). Several groups have reported that levels of PF_1+2_, FDP and D-dimer in plasma of patients with CSU were elevated in association with the severities of clinical symptoms [45,46]. Moreover, the potential of thrombin-producing activity in response to TF exposure in plasma of patients with CSU is elevated, compared to that in the plasma of healthy donors [47]. Active forms of coagulation/fibrinolysis factors not only activate descending factors in the coagulation/fibrinolysis pathway, but may also cleave the extracellular domain of protease-activated receptors (PARs) by serine-specific protease activities. PARs are expressed on the surface of several types of cells, including vascular endothelial cells, monocytes, platelets, monocytes, mast cells, basophils, and neuron, coupled with G-proteins, and mediate information to target cells [48]. To date, four types of mammalian PARs (PAR-1/2/3/4) have been reported. Among coagulation factors, VIIa, Xa and IIa may activate PAR-2, PAP-1,2,3 and PAR-1,3,4, respectively. The complement system is composed of several molecules including complement 1 (C1)-C9. As the coagulation system, most complement components are normally inactive, but sequentially activated via enzymatic cascade in response to the recognition of molecular components of microorganism or non-self-antigens via immunoglobulins. It is classified into three pathways; the classical pathway, the alternative pathway and the lectin pathway [49]. As shown in Figure 2, the classical pathway is triggered by the activation of C1 bound to the immune complex of IgG. The alternative pathway commences with the cleaving of C3 into C3b and C3a (anaphylatoxin). C3b cleaves C5 into C5b and C5a (anaphylatoxin). C5b creates a complex with C6–C9 to make a pore of plasma membrane of microorganisms (host defense). On the other hand, C3a and C5a activate macrophages, neutrophils, basophils and mast cells via C3aR and C5aR, respectively, resulting in allergic reactions. Recently, thrombin (IIa) and plasmin have been reported to contribute to nontraditional complement activation, even in the absence of C3b and C4 [50,51,52]. Thus, the activation of the coagulation cascade may play critical roles for the production of the C3a and C5a by the complement system-independent manner. The other report suggested that C5a levels in plasma is increased in patients with CSU [53].

## 5. The Role of Vascular Endothelial Cells, Eosinophils and Monocytes for the Activation of Coagulation Pathway

As described above, the extrinsic coagulation pathway is usually activated when a blood vessel is damaged and coagulation factors are exposed to TF expressed out of a blood vessel. However, we and other groups have reported that human umbilical vein endothelial cells (HUVECs) and human dermal microvascular endothelial cells (HMVECs) express a large amount of TF on the surface of cells in response to the combination of several molecules (TF-inducers), such as histamine, VEGF, LPS, TNFα, IL-33 and IL-1β, without damage of the blood vessel [54,55]. We divided TF-inducers into two groups according to their signaling pathways, as shown in Figure 2; Group 1 (LPS, TNF-α, IL-33, IL-β) and Group 2 (histamine, VEGF). Factors in Group 1 activate the nuclear factor-kappa B (NF-kB)-related signaling pathway, whereas histamine and VEGF in Group 2 activate the phospholipase C-linked pathway. A high expression of TF on vascular endothelial cells was induced by the co-stimulation of different types of molecules; one in Group 1 and another in Group 2 TF-inducers. Moreover, the highly expressed TF on HUVECs induced the activation of the extrinsic coagulation pathways and produced active forms of the coagulation factor, and then induced the gap-formation of vascular endothelial cells via PAR1, followed by a leakage of plasma from blood vessels (Figure 2) [54,55]. TF expression on vascular endothelial cells induced by TF inducers was suppressed by adenosine: a metabolite of ATP [54]. Of note, adenosine does not only prevent a newly expression of TF, but also decreases TF which is already expressed on endothelial cells on a dose-dependent manner, suggesting that adenosine-related molecules might be effective therapeutically for CSU [54]. As other sources of TF in blood vessel, we found that TF expression levels on human peripheral blood monocytes are significantly increased in CSU patients compared with healthy donors. Moreover, we clarified that stimulation via TLR, -1, -2, -4, or -5 enhanced TF expression levels on the surface of human peripheral monocytes [56]. Moreover, Asero et al. reported that eosinophils express TF on their surface and migrate into the dermis in response to stimulation via the low-affinity IgE receptor, FcɛRII (CD23) [57]. These findings suggest that monocytes and eosinophils also activate the extrinsic coagulation cascade in blood vessels, together with TF expressed on vascular endothelial cells. However, clinical evidence on the roles of TF expressed on vascular endothelial cells, monocytes and eosinophils in patients with CSU is still limited. Further studies on the functions of them, especially in relation to disease severity and reactivities to currently available medications are warranted to find useful drugs and treatments for severe and refractory CSU patients.

## 6. The Role of Basophils as Triggers of CSU 

The role of basophils and mast cells for the development of urticaria through the extrinsic coagulation cascade, protease receptors and complement cascade may be explained by the following model. As described above, several autoantibodies against IgE antibody and/or FcεRI (IgG antibody) and self-molecules, such as IL-24, dsDNA and TPO (IgE antibody), contribute to the basophils and mast cells activation, resulting in the release of chemical mediators, such as histamine, followed by the development of urticaria (autoimmune CSU). In addition, vascular endothelial cells and the extrinsic coagulation pathway may also contribute to the activation of peripheral blood basophils and skin mast cells by the autoimmune system independent pathway. A combination of several exacerbating factors of CSU (TF inducers), such as histamine, VEGF, TNF, IL-1β, IL-33 and LPS, induces high levels of TF expression on vascular endothelial cells, and triggers the extrinsic coagulation pathway [54,55]. Since the combination of histamine and LPS/TNF potently increases the TF expression of vascular endothelial cells, the spontaneous release of histamine from basophils may contribute to the high expression of TF on the surface of vascular endothelial cells. When basophils are activated, PS is also expressed out of cells. This phenomena may accelerate the activation of the extrinsic coagulation cascade. Highly expressed TF on the surface of vascular endothelial cells then activates the extrinsic coagulation pathway together with PS and Ca^2+^ and produces active forms of coagulation factors, such as coagulation factors, Xa and IIa, and the fibrinolysis factor, plasmin. In addition, the high expression of TF was reported on the surface of eosinophils in the lesion of urticaria and monocytes in the peripheral blood of patients with CSU. TF expressed by these cells may also contribute to the activation of the coagulation cascade (Figure 2). Then, Xa and IIa bind to PAR1 on the surface of vascular endothelial cells and induce gap formation and the leakage of plasma from blood vessels. Consequentially, Xa, IIa, and plasmin produces C5a (anaphylatoxin) from the C5 complement by the complement cascade-independent manner, and induces histamine release from basophils and skin mast cells via the C5aR, resulting in the release of a large amount of histamine and the development of urticaria (Figure 1 and Figure 2) [44,54]. It is noteworthy that C5aRs are selectively expressed on the surface of both peripheral basophils and skin mast cells, but not of other kinds of mast cells in humans. Since the release of histamine from human peripheral basophils and skin mast cells in response to C3a is low compared to C5a, C3a may regulate other functions, such as the migration and accumulation of basophils and mast cells to the lesion of urticaria via C3aR [44,54]. 

## 7. Treatment Targeting Basophil-Related Molecules in the Pathogenesis of CSU

In many patients with CSU, peripheral blood basophils and skin mast cells may be activated in response to several types of auto-antibodies (IgG and IgE). Treatment with the anti-IgE antibody quickly decreases serum IgE, and then decreases the expression levels of FcεRI on the surface of peripheral blood basophils followed by those on skin mast cells. Since, FcεRI dependent activation is transduced via several tyrosine kinase, such as Lyn, Syk and Btk, in mast cells and basophils, inhibitors for these molecules should also be useful in the treatment of CSU [58,59,60]. In addition, the involvement of the coagulation-complement cascade in the vascular hyper permeability via Xa- and IIa-PAR1 interaction, and the activation of basophils and mast cells by C5a-C5aR interaction, suggests the efficacy of inhibitors against the coagulation cascade, such as warfarin and heparin, antagonists or antibodies for PAR1, and those against C5a and C5aR for CSU refractory to currently used medications. Moreover, since adenosine suppresses the gap formation of vascular endothelial cells and antigen-IgE-related histamine release from peripheral blood basophils and skin mast cells activation via adenosine receptors (A2a and A2b), adenosine and adenosine analogs (including agonist for A2a and A2b) could also be effective for treating CSU. On the other hand, crucial involvements of IL-4 and IL-5 in the pathogenesis of CSU were suggested via the favorable results of clinical trials of the anti-IL-4/13 receptor antibody, dupilumab [61], anti-IL-5 (mepolizumab, reslizumab) [62,63] and anti-IL-5R (benralizumab) antibody [64]. Moreover, levels of IL-4 in plasma and numbers of IL-4 expressing cells in the skin of patients with CSU are higher than those of health controls [65,66]. IL-5 may also contribute to the pathogenesis of CSU by the activation of eosinophils and basophils [60]. Of note, basophils and eosinophils are major sources of IL-4 and IL-5, respectively, and their decrease in peripheral circulation is a character of sever and type 2b-autoimmune CSU, which is refractory to antihistamines and omalizumab [67].

## 8. Conclusions

In this review, we focused on the role of basophils and introduced details of the behavioral changes of basophils in CSU. These observations show that not only basophil-associated biomarkers, such as plasma histamine and total IgE levels, but also cytomarkers, such as basophil numbers in peripheral blood, the ratio in leukocytes, responsiveness to stimuli, expression of receptors and location may also be useful information to determine the status of patients with CSU. Moreover, we introduced potential roles of basophil- and mast cell- receptors, PARs, adenosine receptors and C5aR. These findings suggest that agonists, antagonists or antibodies against these molecules could be explored as useful drugs for patients with CSU. Further studies on the detailed roles of basophils and the role of their receptors in CSU are warranted to clarify the detailed pathogenesis and discover more useful markers and potential treatments of CSU. 

## Figures and Tables

**Figure 1 ijms-24-10320-f001:**
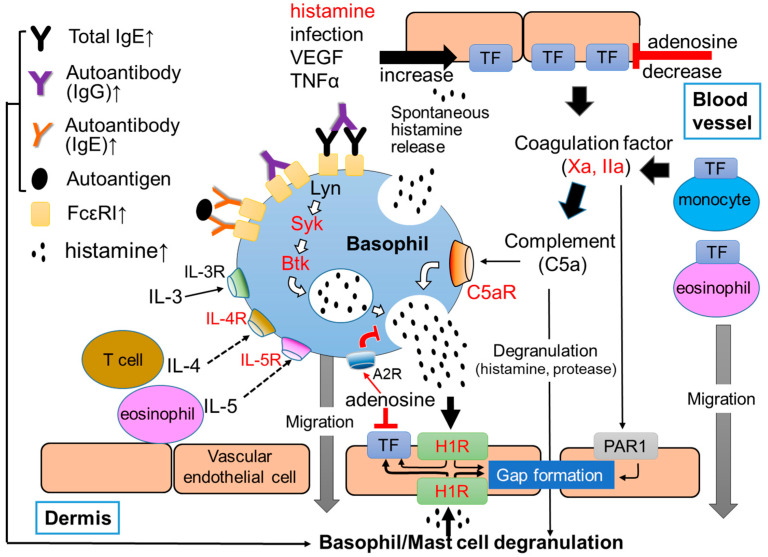
Summarized image of the role of basophils in CSU. Red characters show targets of current and/or developing for CSU. Upward arrows at the end of molecule names indicate the elevation of their concentration in sera of patients with CSU.

**Figure 2 ijms-24-10320-f002:**
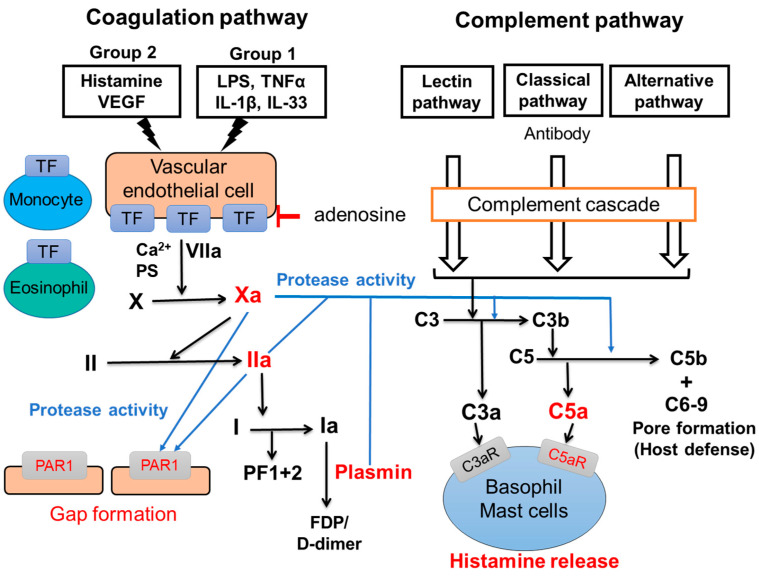
Interactions of coagulation and complement pathway. Blue lines show direct activation of PAR1 and complement pathway-independent production of anaphylatoxins (C3a and C5a) by active forms of coagulation factors. Red characters indicate potential new targets for the treatment for CSU.

## Data Availability

Data presented by this article are available from the corresponding author up on reasonable requests.

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
