# Peer review of "Basophil Characteristics as a Marker of the Pathogenesis of Chronic Spontaneous Urticaria in Relation to the Coagulation and Complement Systems"

_ijms, 2023, doi:10.3390/ijms241210320_

Round 1
Reviewer 1 Report
This is a very comprehensive and detailed review explaining the role of basophils and mast cells in the onset and advancement of chronic spontaneous urticaria. The authors did an extensive literature overview and provided many valuable information regarding the topic. However, I do have some major concerns regarding the paper structure.
1. The manuscript content doesn’t completely match the title. The title suggests basophil characteristics will be analyzed as biomarkers of disease. However, the manuscript is more oriented towards mechanisms of disease rather than biomarker analysis
2. On the other hand, the title suggests the authors will provide “insights for treatment of chronic spontaneous urticaria”, however the section dealing with this issue is insufficiently thorough.
3. The manuscript is too detailed at times, exhaustive and extremely difficult to follow. For example, I find the section “Basophil and mast cell differentiation” completely redundant and unrelated to the subject. Other sections, like “Coagulation system, protease receptors and complement system” and “Basophil-related molecules in plasma and on cell surface of patients with CSU” could use some shortening and narrowing the focus to the subject.
4. Overall, I suggest the authors rewrite mentioned parts of the manuscript in a way to provide less unimportant details and more focus, as to make it more interesting to the readers and easier to follow.
Author Response
Answers to reviewer 1
This is a very comprehensive and detailed review explaining the role of basophils and mast cells in the onset and advancement of chronic spontaneous urticaria. The authors did an extensive literature overview and provided many valuable information regarding the topic. However, I do have some major concerns regarding the paper structure.
- The manuscript content doesn’t completely match the title. The title suggests basophil characteristics will be analyzed as biomarkers of disease. However, the manuscript is more oriented towards mechanisms of disease rather than biomarker analysis
Thank you for your important comments. According to your suggestion, we have modified the title of the manuscript so as to cover all topics.
- On the other hand, the title suggests the authors will provide “insights for treatment of chronic spontaneous urticaria”, however the section dealing with this issue is insufficiently thorough.
According to your suggestion, we have modified the title of the manuscript to align with the topics.
- The manuscript is too detailed at times, exhaustive and extremely difficult to follow. For example, I find the section “Basophil and mast cell differentiation” completely redundant and unrelated to the subject. Other sections, like “Coagulation system, protease receptors and complement system” and “Basophil-related molecules in plasma and on cell surface of patients with CSU” could use some shortening and narrowing the focus to the subject.
Thank you for your pointing this out. Based on your comments, we have merged the section of "Basophil and mast cell differentiation" and "Characteristics of basophils and mast cells", and made the comparison of basophils and mast cells clearer (red text). In order to understand the role of basophils in CSU, “Coagulation system, protease receptors and complement system” and “Basophil-related molecules in plasma and on cell surface of patients with CSU” sections are important. Moreover, reviewer 2 recognized the significance of these sections. Therefore, we performed a little modification of these sections.
- Overall, I suggest the authors rewrite mentioned parts of the manuscript in a way to provide less unimportant details and more focus, as to make it more interesting to the readers and easier to follow.
Thank you for your comments. In order to make readers easy to understand the contents of this manuscript, we have modified the title and several sections (red characters).

Reviewer 2 Report
The Authors have compiled a really nice review article on the role basophils in CSU. The manuscript is well-written and flows nicely. Figure 1 is particularly illustrative and adds to the overall value of the paper.
Minor comments, remarks and suggestions:
- Concerning Table 1: please correct "conditons" in "conditions".
- On lines 53-54 consider citing also another important article on the role of IgG and IgG autoantibodies directed against IgE receptors in CSU: Maronese CA, Ferrucci SM, Moltrasio C, et al. IgG and IgE Autoantibodies to IgE Receptors in Chronic Spontaneous Urticaria and Their Role in the Response to Omalizumab. J Clin Med. 2023;12(1):378. doi:10.3390/jcm12010378
Author Response
Answers to reviewer 2
The Authors have compiled a really nice review article on the role basophils in CSU. The manuscript is well-written and flows nicely. Figure 1 is particularly illustrative and adds to the overall value of the paper.
Thank you for your kind comments.
Minor comments, remarks and suggestions:
- Concerning Table 1: please correct "conditons" in "conditions".
Thank you for your pointing it out. We have corrected it.
- On lines 53-54 consider citing also another important article on the role of IgG and IgG autoantibodies directed against IgE receptors in CSU: Maronese CA, Ferrucci SM, Moltrasio C, et al. IgG and IgE Autoantibodies to IgE Receptors in Chronic Spontaneous Urticaria and Their Role in the Response to Omalizumab. J Clin Med. 2023;12(1):378. doi:10.3390/jcm12010378
Thank you for your useful suggestion. We have added the paper as a reference.

Round 2
Reviewer 1 Report
The authors substantially improved the manuscript, therefore, I recommend the paper for publication
Author Response
Thank you so much for your valuable comments.